# Alpha-Fetoprotein Combined with Radiographic Tumor Burden Score to Predict Overall Survival after Liver Resection in Hepatocellular Carcinoma

**DOI:** 10.3390/cancers15041203

**Published:** 2023-02-14

**Authors:** Yi-Hao Yen, Yueh-Wei Liu, Wei-Feng Li, Chih-Chi Wang, Chee-Chien Yong, Chih-Che Lin, Chih-Yun Lin

**Affiliations:** 1Division of Hepatogastroenterology, Department of Internal Medicine, Kaohsiung Chang Gung Memorial Hospital, College of Medicine, Chang Gung University, 123 Ta Pei Road, Kaohsiung 833401, Taiwan; 2Liver Transplantation Center, Department of Surgery, Kaohsiung Chang Gung Memorial Hospital, 123 Ta Pei Road, Kaohsiung 833401, Taiwan; 3Biostatistics Center of Kaohsiung Chang Gung Memorial Hospital, Kaohsiung 833401, Taiwan

**Keywords:** tumor burden score, hepatocellular carcinoma, alpha-fetoprotein, liver resection

## Abstract

**Simple Summary:**

The tumor burden score (TBS) is calculated using the Pythagorean theorem based on the largest tumor size and tumor number (α^2^ + β^2^ = γ^2^, where α is the largest tumor size, β is the tumor number, and γ is the TBS). Patients who underwent liver resection (LR) for Barcelona Clinic Liver Cancer stage 0, A, or B hepatocellular carcinoma (HCC) between 2011 and 2018 were enrolled. Among 743 patients, 193 (26.0%) patients had a low TBS (<2.6), 474 (63.8%) had a moderate TBS (2.6–7.9), and 75 (10.1%) had a high TBS (>7.9). Combining radiographic TBS and alpha-fetoprotein levels could stratify overall survival among HCC patients after LR.

**Abstract:**

We evaluated whether combining the radiographic tumor burden score (TBS) and alpha-fetoprotein (AFP) level could be used to stratify overall survival (OS) among hepatocellular carcinoma (HCC) patients after liver resection (LR). Patients who underwent LR for Barcelona Clinic Liver Cancer stage 0, A, or B HCC between 2011 and 2018 were enrolled. TBS scores were calculated using the following equation: TBS^2^ = (largest tumor size (in cm))^2^ + (tumor number)^2^. Among 743 patients, 193 (26.0%) patients had a low TBS (<2.6), 474 (63.8%) had a moderate TBS (2.6–7.9), and 75 (10.1%) had a high TBS (>7.9). Those with a TBS ≤ 7.9 and AFP < 400 ng/mL had a significantly better OS than those with a TBS > 7.9 and an AFP < 400 ng/mL (*p* = 0.003) or ≥ 400 ng/mL (*p* < 0.001). A multivariate analysis using TBS ≤ 7.9 and AFP < 400 ng/mL as the reference values showed that a TBS > 7.9 and an AFP < 400 ng/mL (hazard ratio (HR): 2.063; 95% confidence interval [CI]: 1.175–3.623; *p* = 0.012) or ≥ 400 ng/mL (HR: 6.570; 95% CI: 3.684–11.719; *p* < 0.001) were independent predictors of OS. In conclusion, combining radiographic TBSs and AFP levels could stratify OS among HCC patients undergoing LR.

## 1. Introduction

Hepatocellular carcinoma (HCC) is one of the leading causes of cancer-related deaths worldwide [1,2,3]. Liver resection (LR) is a primary treatment modality for HCC [1,2,3] and has been shown to improve survival across Barcelona Clinic Liver Cancer (BCLC) stages [4,5]. However, LR is also associated with higher risks compared with non-surgical treatment [2,3]. Therefore, preoperative prognostic predictions and the assessment of risk-to-benefit ratios in patients with HCC is critical when determining which patients should undergo LR.

A previous study established the tumor burden score (TBS), which is calculated using the largest tumor size and tumor number in the Pythagorean theorem (α^2^ + β^2^ = γ^2^, where α = largest tumor size, β = tumor number, and γ = TBS) [6]. The TBS showed a satisfactory ability to stratify prognostic outcomes among patients with HCC who underwent LR [6]. Further, Tsilimigras et al. reported that serum alpha-fetoprotein (AFP) and pathological TBS had synergistic impacts on prognosis following LR for HCC [7]. However, Tsilimigras et al. used pathological TBS, which is not suitable for preoperative prognostic predictions [7]. We aimed to use radiographic TBS combined with AFP evaluation to predict prognosis in patients with BCLC stages 0, A, and B HCC who underwent LR.

## 2. Materials and Methods

The study was conducted according to the guidelines of the Declaration of Helsinki and approved by the Institutional Review Board of Chang Gung Memorial Hospital—Kaohsiung branch (reference number: 202201189B0; approval date: 8 August 2022).

Data were extracted from the Kaohsiung Chang Gung Memorial Hospital HCC registry data. These data are prospectively collected and updated annually. The vital status of every patient is updated annually through a link to the website of the Taiwan Ministry of Health and Welfare (Cancer Screening and Tracing Information Integrated System for Taiwan; https://hosplab.hpa.gov.tw/CSTIIS/index.aspx, accessed on 1 January 2007).

From 2011 to 2018, patients with newly diagnosed HCC who underwent LR at Kaohsiung Chang Gung Memorial Hospital were enrolled in this study. The inclusion criteria were BCLC stage 0, A, and B HCC. The exclusion criteria were unknown preoperative AFP level, unknown tumor differentiation status, age < 18 years, unknown pathological stage, pathological stage N1 or M1, and non-curative LR. Curative LR was defined as the complete resection of all macroscopic tumors with microscopically negative surgical margins. After the application of all inclusion and exclusion criteria, 743 patients who underwent LR during 2011–2018 were enrolled. The raw data for all 743 included patients are available via the following digital object identifier: https://www.dropbox.com/scl/fi/unezavq59ethkc6sprtu9/raw-data.xlsx?dl=0&rlkey=cwu1zq6f9bguxiq0ghqqfvvh3, accessed on 1 January 2023.

Tumor sizes and numbers were assessed using preoperative contrast-enhanced computed tomography or magnetic resonance imaging. Tumor differentiation was assessed using Edmondson and Steiner’s classification [8]. Fibrosis was assessed using Ishak scores [9]. Cirrhosis was defined as an Ishak score of 5 or 6. Major resection was defined as the resection of ≥3 Couinaud segments.

The Seventh American Joint Committee on Cancer (AJCC)/tumor–node–metastasis (TNM) staging criteria [10] were applied to our HCC registry data from 2011 to 2017, and the Eighth AJCC/TNM staging criteria [11] were applied to our HCC registry data starting in 2018. Therefore, we present the pathological T-stage in this study as stage 1 or 2 versus stage 3 or 4. A T-stage 1 or 2 includes the detection of a single tumor, with or without vascular invasion, or multiple tumors, none of which are >5 cm. T-stages 3 or 4 include the detection of multiple tumors, among which any are >5 cm, or tumors with major vascular invasion, the perforation of the visceral peritoneum, or the direct invasion of adjacent organs other than the gallbladder.

## 3. Outcome Measurement

The primary outcome measure was overall survival (OS), which was defined as the interval between the date of LR and the date of the last follow-up or death.

## 4. Definitions

The detection of a single tumor ≤ 2 cm was defined as BCLC stage 0; a single tumor > 2 cm, or 2–3 tumors ≤ 3 cm, were defined as BCLC stage A; and 2–3 tumors ≥ 3 cm, or ≥ 4 tumors, were defined as BCLC stage B [5]. Tumor sizes were defined by the size of the largest tumor if multiple tumors were detected. TBS scores [6] were determined using the following equation: TBS^2^ = (largest tumor size (in cm))^2^ + (tumor number)^2^. The tumor number and the largest tumor size were determined based on the results of imaging studies. Cutoff values for TBS were determined according to OS by using X-tile [12], a bioinformatics tool created by Camp et al.

## 5. Statistical Analysis

The patients’ characteristics are presented as the number (percentage) of patients matching the specific characteristics, and they were compared using the Chi-square test. Comparisons of OS between groups were performed using the Kaplan–Meier survival curves and the log-rank test. Covariates in the multivariate model were chosen a priori, based on established clinical relevance. The potential confounders included age, the presence of cirrhosis, TBS, AFP level, and tumor differentiation status [2,3,5,6]. These variables were fully adjusted in the multivariate model. The results are presented as a hazard ratio (HR) with a 95% confidence interval (CI). Statistical analyses were performed using SPSS version 22.0. Two-tailed significance values were calculated, and significance was defined as *p* < 0.05.

## 6. Results

### 6.1. Clinical and Pathological Characteristics of Patients

A total of 743 patients, who underwent LR for HCC and met the inclusion and exclusion criteria, were enrolled in this study. The patients were divided into three groups according to the TBS: high TBS (>7.9; *n* = 75, 10.1%), medium TBS (2.6–7.9; *n* = 474, 63.8%), and low TBS (<2.6; *n* = 193, 26%) (Figure 1). Overall, 136 patients (18.3%) had BCLC stage 0, 538 (72.4%) had BCLC stage A, and 69 (9.3%) had BCLC stage B. Among all 743 patients, 390 (52.5%) patients were hepatitis B surface antigen (HBsAg)-positive, 251 (33.8%) patients were anti-hepatitis C virus (HCV)-positive, 352 (47.4%) patients had a major resection, 414 (55.7%) patients were < 65 years of age, 572 (77.0%) patients were male, 623 (83.8%) patients had AFP < 400 ng/mL, and 731 (98.4%) patients were Child–Pugh Class A. Pathological examinations showed that 287 (38.6%) patients had cirrhosis, 29 (3.9%) patients had poorly differentiated tumors, and 63 (8.5%) patients had pathological T-stage 3 or 4 (Table 1). After combining TBS and AFP levels, four groups were generated: TBS ≤ 7.9/AFP < 400 ng/mL (*n* = 577, 77.7%); TBS ≤ 7.9/AFP ≥ 400 ng/mL (*n* = 90, 12.1%); TBS > 7.9/AFP < 400 ng/mL (*n* = 46, 6.2%); and TBS > 7.9/AFP ≥ 400 ng/mL (*n* = 30, 4.0%). The proportion of male patients (89.1%) was highest among the TBS > 7.9/AFP < 400 ng/mL group (*p* = 0.014). The proportion of cirrhotic patients (15.2%) was lowest in the TBS > 7.9/AFP ≤ 400 ng/mL group (*p* = 0.014). The proportion of patients who were anti-HCV-positive (36.4%) was highest in the TBS ≤ 7.9/AFP < 400 ng/mL group (*p* = 0.023). The proportion of patients who underwent major resection (86.7%) was highest in the TBS > 7.9/AFP ≥ 400 ng/mL group (*p* < 0.001). The proportion of patients with poor tumor differentiation (13.3%) was highest in the TBS > 7.9/AFP ≥ 400 ng/mL group (*p* < 0.001). The proportion of patients with pathological stage-T 3 or 4 (60.0%) was highest in the TBS > 7.9/AFP ≥ 400 ng/mL group (*p* < 0.001). No significant differences between groups were observed for the remaining variables (Table 2).

### 6.2. The Association of TBS and AFP with OS

The median follow-up in this cohort was 19.9 months (interquartile range (IQR): 10.8–58.8 months). Both the TBS and AFP were strong predictors of OS, and the 5-year OS incrementally worsened with higher TBS (low TBS: 86%; medium TBS: 69%; high TBS: 44%; *p* < 0.001) (Figure 1). Patients with high AFP levels (i.e., AFP ≥ 400 ng/mL) had worse 5-year OS compared with patients who had low AFP levels (i.e., AFP < 400 ng/mL; 57% vs. 73%; *p* < 0.001).

When examining different combinations of TBS and AFP relative to OS, TBS and AFP showed a synergistic impact on OS. Patients with TBS ≤ 7.9 and AFP < 400 ng/mL had better OS than those with TBS > 7.9 and AFP < 400 ng/mL (*p* = 0.003) and those with TBS > 7.9 and AFP ≥ 400 ng/mL (*p* < 0.001). Patients with TBS ≤ 7.9 and AFP ≥ 400 ng/mL had better OS than those with TBS > 7.9 and AFP ≥ 400 ng/mL (*p* < 0.001). Patients with TBS > 7.9 and AFP < 400 ng/mL had better OS than those with TBS > 7.9 and AFP ≥ 400 ng/mL (*p* = 0.027). No significant differences in OS were observed between other groups (Figure 2).

On multivariate analysis, using TBS ≤ 7.9 and AFP < 400 ng/mL as the reference values, and after adjusting for all confounding variables, the combinations of TBS > 7.9/AFP < 400 ng/mL (HR: 2.063; 95% CI: 1.175–3.623; *p* = 0.012) and TBS > 7.9/AFP ≥ 400 ng/mL (HR: 6.57; 95% CI: 3.684–11.719; *p* < 0.001) were independent predictors of OS. By contrast, the combination of TBS ≤ 7.9/AFP ≥ 400 ng/mL was not an independent predictor for OS (HR: 1.394; 95% CI: 0.858–2.266; *p* = 0.180; Table 3).

## 7. Discussion

In the current study, TBS combined with AFP levels could stratify OS among patients with BCLC stage 0, A, and B HCC who underwent LR. This model was useful for preoperative prognosis prediction and the assessment of risk-to-benefit ratios. The proportions of cirrhotic patients were lowest in the TBS > 7.9/AFP < 400 ng/mL group, which also contained higher proportions of patients with high tumor burden who required major resection. Major resection can be performed in patients with non-cirrhotic HCC with low rates of major complications and with satisfactory outcomes [13]. The proportion of patients with poor tumor differentiation and pathological T-stage 3 or 4 was highest in the TBS > 7.9/AFP ≥ 400 ng/mL group, indicating a high tumor burden, and a previous study reported that AFP elevation was correlated with vascular invasion and poor tumor differentiation [14].

The BCLC guidelines suggest that LR is indicated for BCLC stages 0–A [5]. However, a meta-analysis reported that LR improved the prognosis of BCLC B HCC patients compared with those who underwent transarterial chemoembolization (TACE) [15]. Furthermore, LR for BCLC B HCC is commonly adopted in daily practice at both Eastern and Western treatment centers [16,17]. We enrolled BCLC stage 0, A, and B patients in the current study.

BCLC stages A and B are heterogeneous. BCLC stage A is defined as multiple tumors within the Milan criteria or a solitary tumor >2 cm, irrespective of the size [5]. However, increasing tumor size is associated with an increased risk of microvascular invasion and micrometastasis, which can lead to a worse prognosis [18,19,20]. A previous study reported that the prognosis of BCLC stage A patients with a single large HCC > 5 cm is similar to that of patients classified as BCLC stage B HCC among patients who underwent LR [21]. BCLC stage B is well-known for its heterogeneity [3]. The BCLC staging system does not consider AFP in their model [5], despite AFP being a well-known prognostic biomarker for HCC [22]. We, therefore, hypothesized that TBS combined with AFP evaluation could be useful in daily practice for predicting OS among patients with BCLC stage 0, A, and B HCC who undergo LR.

The optimal cutoff values for TBS relative to OS were determined in the current study using the X-tile bioinformatics tool [12] (low TBS was defined as < 2.6, medium TBS was defined as 2.6–7.9, and high TBS was defined as > 7.9); these values are different from those reported by Tsilimigras et al. (low TBS was defined as < 3.36, medium TBS was defined as 3.36–13.74, and high TBS was defined as > 13.74) [7]. The discrepancies between studies could be due to differences in patients’ characteristics. Furthermore, Tsilimigras et al. used pathologically defined TBS [7], whereas we used radiographically defined TBS in the current study.

The TBS is simple to use, and numerous studies have demonstrated its predictive value in prognosis in patients with HCC undergoing LR [6,7,23,24,25,26,27,28,29,30,31,32]. Among these studies, some studies used pathological TBS, which is not suitable for preoperative prognostic predictions [6,7,23,24,25,26]. Two studies only enrolled patients with BCLC stage B [27,28]. Whether the results of these studies could be extrapolated to other BCLC stages is unknown; meanwhile, our study enrolled patients with BCLC stages 0, A, and B. Endo et al. reported that a preoperative model composed of AFP, radiographic TBS, and neutrophil-to-lymphocyte ratio could predict the presence of microvascular invasion [29], whereas the aim of the present study is to predict OS. Fukami et al. enrolled patients with BCLC stages 0, A, and B. The authors used a combination of the Controlling Nutritional Status score and radiographic TBS to predict OS. However, this study only enrolled 96 patients. The case number may be too small to draw any conclusions [30].

Endo et al. enrolled 1676 patients with BCLC stages 0, A, B, and C. The authors used a preoperative model composed of radiographic TBS, AFP, neutrophil-to-lymphocyte ratio, albumin, gamma-glutamyl transpeptidase, and vascular involvement to predict 5-year OS. This model could stratify the OS into three distinct groups. Further, this model outperformed HCC staging systems including the BCLC and the AJCC systems [31]. Lima et al. enrolled 1435 patients with HCC undergoing LR without mentioning their BCLC stage. A risk score that included three variables (i.e., radiographic TBS, AFP, and Child–Pugh class) demonstrated superior predictive value for OS compared with the BCLC stage and further stratified patients within the BCLC stage relative to OS [32].

The advantage of our model, which is composed of radiographic TBS (cutoff value = 7.9) and AFP (cutoff value = 400 ng/mL), is simplicity; meanwhile, Lima et al. used a point system (TBS low/medium/high = 0/1/2; AFP low/high = 0/1; Child–Pugh class A/B = 0/1) to develop a risk score [32] and Endo et al. used an online calculator (https://yutaka-endo.shinyapps.io/PrepoScore_Shiny/, accessed on 1 January 2023) for their complex model [31]. We believe that model simplicity is of paramount importance in clinical application. Although numerous prognostic models have been developed for HCC [33], the BCLC [5] and the AJCC [10,11] staging systems are the most popular, mainly due to their simplicity.

One strength of the current study was the evaluation of radiographic TBS, which is useful for preoperative risk assessment. In addition, the vital status of each patient in the current study was verified through a link to a government website. There were few pieces of missing data identified in the current study. Among the variables included in the multivariate analysis, only 36 (4.8%) patients had unknown cirrhotic status. The limitations of the current study include the retrospective design and the study of a single treatment center, the outcomes of which may not be generalizable to other institutions. Furthermore, the majority of patients in this study were categorized as low tumor burden (i.e., TBS ≤ 7.9 and AFP < 400 ng/mL). The case numbers in the remaining groups were limited. Finally, no validation cohort was used to verify the cutoff values for TBS.

## 8. Conclusions

Radiographic TBS combined with AFP levels successfully stratified OS among patients with BCLC stages 0, A, and B in patients who underwent LR, which could be used for the preoperative prognosis prediction and risk-to-benefit ratio assessments among patients with BCLC stage 0, A, and B HCC prior to LR.

## Figures and Tables

**Figure 1 cancers-15-01203-f001:**
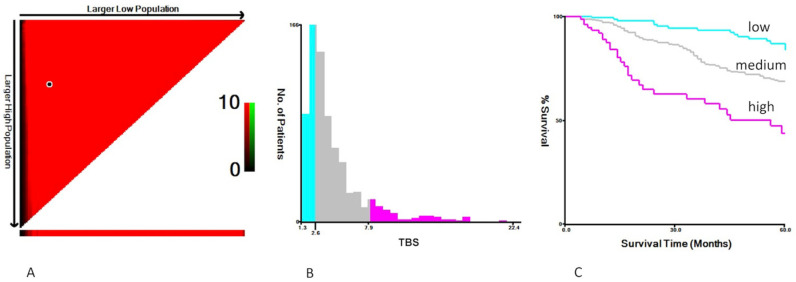
Cutoff tumor burden score (TBS) values were determined by overall survival using X-tile, a bioinformatics tool created by Camp et al. (**A**) Data represented graphically in a right-triangular grid in which each pixel represents the data from a given set of divisions. The vertical axis represents all possible “high” populations, with the size of the high population increasing from top to bottom. Similarly, the horizontal axis represents all possible “low” populations, with the size of the low population increasing from left to right. (**B**) The number of patients in each group for a given set of divisions. (**C**) Kaplan–Meier curves show significant differences in overall survival among patients with a low TBS (i.e., <2.6), moderate TBS (i.e., 2.6–7.9), and high TBS (i.e., >7.9; *p* < 0.001).

**Figure 2 cancers-15-01203-f002:**
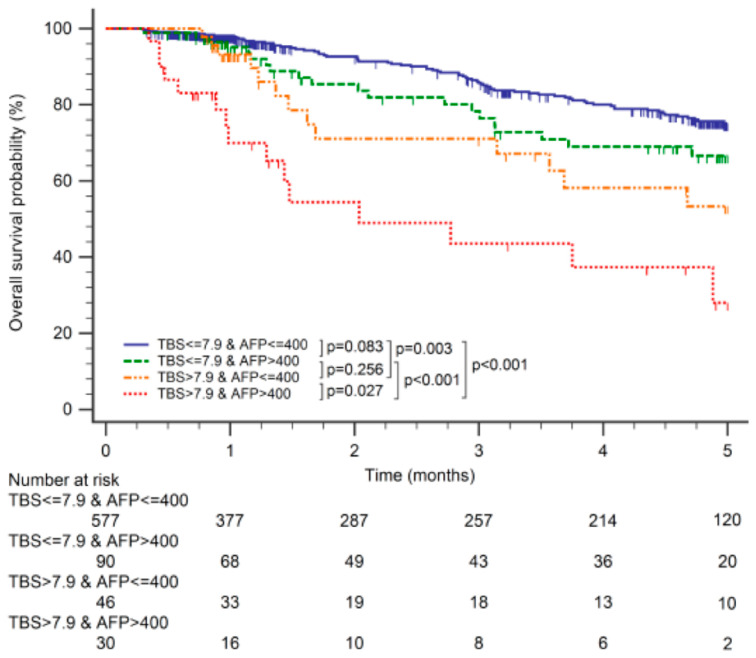
Kaplan–Meier curves demonstrating differences in overall survival among patients with tumor burden score (TBS) ≤ 7.9/α-fetoprotein (AFP) ≤ 400 ng/mL, TBS ≤ 7.9/AFP > 400 ng/mL, TBS > 7.9/AFP ≤ 400 ng/mL, and TBS > 7.9/AFP > 400 ng/mL.

**Table 1 cancers-15-01203-t001:** The patients’ clinical and pathological characteristics.

	Total, *n* = 743
Age (year)	
≤65	414 (55.7%)
>65	329 (44.3%)
Sex	
Male	572 (77.0%)
Female	171 (23.0%)
Cirrhosis	
Presence	287 (38.6%)
Absence	420 (56.5%)
Unknown	36 (4.8%)
HBsAg	
Positive	390 (52.5%)
Negative	353 (47.5%)
Anti-HCV	
Positive	251 (33.8%)
Negative	492 (66.2%)
AFP (ng/mL)	
<400	623 (83.8%)
≥400	120 (16.2%)
Child–Pugh class	
A	731 (98.4%)
B	11 (1.5%)
Unknown	1 (0.1%)
Type of resection	
Major	352 (47.4%)
Minor	394 (53.0)
Tumor differentiation	
Poor	29 (3.9%)
Moderate	658 (88.6%)
Well	56 (7.5%)
Pathology T-stage	
1–2	680 (91.5%)
3–4	63 (8.5%)
Tumor burden score	
Low < 2.6	193 (26.0%)
Medium: 2.6–7.9	474 (63.8%)
High > 7.9	75 (10.1%)
BCLC stage	
0	136 (18.3)
A	538 (72.4%)
B	69 (9.3%)
Radiographic tumor size (cm)	
≤5	573 (77.1%)
>5	170 (22.9%)
Radiographic tumor number	
1	637 (85.7%)
2	78 (10.5%)
3	16 (2.2%)
4	11 (1.5%)
5	1 (0.1%)

BCLC, Barcelona Clinic Liver Cancer; HBsAg, hepatitis B virus surface antigen; HCV, hepatitis C virus; AFP, α-fetoprotein.

**Table 2 cancers-15-01203-t002:** Clinical and pathological characteristics of patients stratified by AFP and TBS levels.

	TBS ≤ 7.9 and AFP < 400 ng/mL, *n* = 577	TBS ≤ 7.9 and AFP ≥ 400 ng/mL, *n* = 90	TBS > 7.9 and AFP < 400 ng/mL, *n* = 46	TBS > 7.9 and AFP ≥ 400 ng/mL, *n* = 30	*p*
Age (year)					0.283
≤65	381 (66%)	58 (64.4%)	27 (58.7%)	24 (80%)	
>65	196 (34%)	32 (35.6%)	19 (41.3%)	6 (20%)	
Sex					0.014
Male	488 (77.6%)	59 (65.6%)	41 (89.1%)	23 (76.7%)	
Female	129 (22.4%)	31 (34.4%)	5 (10.9%)	7 (23.3%)	
Cirrhosis					<0.001
Yes	231 (40%)	43 (47.8%)	7 (15.2%)	6 (20%)	
No	323 (56%)	43 (47.8%)	33 (71.7%)	21 (70%)	
Unknown	23 (4.0%)	4 (4.4%)	6 (13.0%)	3 (10.0%)	
HBsAg					0.076
Positive	302 (52.3%)	52 (57.8%)	17 (37%)	19 (36.7%)	
Negative	275 (47.7%)	38 (42.2%)	29 (63.0%)	11 (63.3%)	
Anti-HCV					0.023
Positive	210 (36.4%)	26 (28.9%)	12 (26.1%)	4 (13.3%)	
Negative	367 (63.6%)	64 (71.1%)	34 (73.9%)	26 (86.7%)	
Child–Pugh class					0.168
A	568 (98.4%)	90 (100%)	43 (93.5%)	30 (100%)	
B	8 (1.4%)	0	3 (6.5%)	0	
Unknown	1 (0.2%)	0	0	0	
Type of resection					<0.001
Major	245 (42.5%)	41 (42.6%)	37 (80.4%)	26 (86.7%)	
Minor	332 (57.5%)	49 (54.4%)	9 (19.6%)	4 (13.3%)	
Tumor differentiation					<0.001
Poor	14 (2.4%)	8 (8.9%)	3 (6.5%)	4 (13.3%)	
Moderate	510 (88.4%)	81 (90.0%)	42 (91.3%)	24 (83.3%)	
Well	53 (9.2%)	1 (1.1%)	1 (2.2%)	1 (3.3%)	
Pathology T-stage					<0.001
1–2	551 (95.5%)	84 (93.3%)	33 (71.7%)	12 (40%)	
3–4	26 (4.5%)	6 (6.7%)	13 (28.3%)	18 (60%)	

HBsAg, hepatitis B virus surface antigen; HCV, hepatitis C virus; AFP, α-fetoprotein; TBS, tumor burden score.

**Table 3 cancers-15-01203-t003:** Univariate and multivariate Cox regression analysis of factors associated with overall survival.

	Univariate		Multivariate	
	HR (95% CI)	*p*	HR (95% CI)	*p*
Age (year)				
≤65	1.00 (reference)		1.00 (reference)	
>65	2.083 (1.488–2.915)	<0.001	2.146 (1.523–3.024)	<0.001
Cirrhosis				
No	1.00 (reference)		1.00 (reference)	
Yes	1.525 (1.081–2.153)	0.016	1.692 (1.190–2.405)	0.003
Unknown	1.382 (0.597–3.197)	0.450	0.969 (0.414–2.268)	0.942
Tumor differentiation				
Well or moderate	1.00 (reference)		1.00 (reference)	
Poor	2.895 (1.514–5.535)	0.001	2.027 (1.044–3.939)	0.037
Group				
TBS ≤ 7.9 and AFP < 400 ng/mL	1.00 (reference)		1.00 (reference)	
TBS ≤ 7.9 and AFP ≥ 400 ng/mL	1.560 (0.966–2.520)	0.069	1.394 (0.858–2.266)	0.180
TBS > 7.9 and AFP < 400 ng/mL	2.077 (1.193–3.617)	0.010	2.063 (1.175–3.623)	0.012
TBS > 7.9 and AFP ≥ 400 ng/mL	5.378 (3.094–9.349)	<0.001	6.57 (3.684–11.719)	<0.001

AFP, α-fetoprotein; TBS, tumor burden score.

## Data Availability

The data presented in this study are available via the following digital object identifier: https://www.dropbox.com/scl/fi/unezavq59ethkc6sprtu9/raw-data.xlsx?dl=0&rlkey=cwu1zq6f9bguxiq0ghqqfvvh3, accessed on 1 January 2023.

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
