# Peer review of "Alpha-Fetoprotein Combined with Radiographic Tumor Burden Score to Predict Overall Survival after Liver Resection in Hepatocellular Carcinoma"

_cancers, 2023, doi:10.3390/cancers15041203_

Round 1

Reviewer 1 Report

Congratulations for your work to predict survival using TBS and AFP.

TBS and AFP combined are well known to predict both microvascular invasion and survival. Unfortunately, similar articles are already published (Cancers (Basel). 2021 Feb 11;13(4):747  Cancers (Basel). 2021 Aug 31;13(17):4403 J Gastrointest Surg. 2022 Dec;26(12):2512-2521). 

I cannot find sufficient evidence for cut-off value of TBS and moreover AFP (400). AFP 400 is too high and it is not surprising that there is survival difference between groups.

Author Response

Congratulations for your work to predict survival using TBS and AFP.

TBS and AFP combined are well known to predict both microvascular invasion and survival. Unfortunately, similar articles are already published (Cancers (Basel). 2021 Feb 11;13(4):747  Cancers (Basel). 2021 Aug 31;13(17):4403 J Gastrointest Surg. 2022 Dec;26(12):2512-2521).

Response: Thank you so much for your comments.

The present study found that the combination of radiographic TBS and AFP levels could be used to stratify overall survival (OS) among patients with BCLC Stages 0, A, and B HCC who underwent liver resection (LR).

Tsilimigras et al. reported that serum AFP and pathological TBS had synergistic impacts on prognosis following LR for HCC. However, Tsilimigras et al. used pathological TBS, which is not suitable for preoperative prognostic predictions. (Cancers (Basel). 2021 Feb 11;13(4):747)

Lee et al, reported that preoperative total tumor volume (TTV) and AFP to be independent risk factors for microvascular invasion. The TTV was the sum of the volume of every tumor, which was calculated as: length × width × width × 0.52 (Cancers (Basel). 2021 Aug 31;13(17):4403). Whereas the tumor burden score (TBS), which is calculated using the largest tumor size and tumor number in the Pythagorean theorem (α2 + β2 = γ2, where α = largest tumor size, β = tumor number, and γ= TBS).

Lima et al. enrolled 194 patients with BCLC stage B HCC underwent LR. They found that AFP combined radiographic TBS could predict recurrence-free survival and OS. (J Gastrointest Surg. 2022 Dec;26(12):2512-2521). Whereas the present study enrolled patients with BCLC Stages 0, A, and B HCC who underwent LR.

I cannot find sufficient evidence for cut-off value of TBS and moreover AFP (400). AFP 400 is too high and it is not surprising that there is survival difference between groups.

Response: 

  1. Cutoff values for TBS were determined according to overall survival (OS) by using X-tile, a bioinformatics tool created by Camp et al (Clin Cancer Res 2004; 10: 7252–7259). The patients were divided into three groups according to the TBS: high TBS (>7.9; n = 75, 10.1 %), medium TBS (2.6–7.9; n = 474, 63.8 %), and low TBS (<2.6; n = 193, 26%) (Fig.1). Both TBS and AFP were strong predictors of OS, and 5-year OS incrementally worsened with higher TBS (low TBS: 86%; medium TBS: 69%; high TBS: 44%; p < 0.001) (Fig.1). Patients with high AFP levels (i.e., AFP ≧400 ng/mL) had worse 5-year OS compared with patients with had low AFP levels (i.e., AFP < 400 ng/mL; 57% vs. 73%; p < 0.001). Please see line 161-165.
  2. We used 400 ng/ml as the cutoff value of AFP in accordance with the Cancer of the Liver Italian Program (CLIP) staging system. (Hepatology 1998;28:751-5.)

Reviewer 2 Report

The manuscript entitled “Alpha-Fetoprotein Combined with Radiographic Tumor Burden Score to Predict Overall Survival After Liver Resection in Hepatocellular Carcinoma” by Yen and coworkers addresses a interesting. Overall, the manuscript is well written, sound and the data is concise. However, a few points might be addressed:

1.)    Very recently, a manuscript by Lima by et al. addresses the similar point using the radiographic TBS and an AFP cut-off at 400 ng/ml. The authors should point out what is the difference to their study and what are a novelties in their analysis.

2.) What is the rational for using the Up-to-7 criteria in combination with AFP in the context of this analysis?

3.) The results from Up-to-7 criteria in addition to AFP levels with regard to the prognostic stratification is not even mentioned in the abstract although the shown data is interesting.  

Author Response

The manuscript entitled “Alpha-Fetoprotein Combined with Radiographic Tumor Burden Score to Predict Overall Survival After Liver Resection in Hepatocellular Carcinoma” by Yen and coworkers addresses a interesting. Overall, the manuscript is well written, sound and the data is concise. However, a few points might be addressed:

  • Very recently, a manuscript by Lima by et al. addresses the similar point using the radiographic TBS and an AFP cut-off at 400 ng/ml . The authors should point out what is the difference to their study and what are a novelties in their analysis.

Response: Thank you so much for your comments. The present study enrolled 743 patients who underwent liver resection (LR) for BCLC stage 0, A, or B HCC during 2011–2018 in our institution. After combining TBS and AFP levels, four groups were generated: TBS ≤ 7.9/AFP < 400 ng/ml (n = 577, 77.7%); TBS ≤ 7.9/AFP ≧400 ng/ml (n = 90, 12.1%); TBS > 7.9/AFP <400 ng/ml (n = 46, 6.2%); and TBS > 7.9/AFP ≧400 ng/ml (n = 30, 4.0%). The combination of radiographic TBS and AFP levels could be used to stratify OS among HCC patients who undergo LR.

Lima et al. enrolled 1435 patients who underwent curative‐intent resection for HCC from a multi‐institutional database. A risk score which included three variables (i.e., radiographic TBS, AFP and Child-Pugh class) demonstrated superior predictive value for prognosis compared with BCLC stage and further stratified patients within BCLC groups relative to OS. The authors concluded that their risk score is a simple, holistic score that consistently outperformed BCLC stage relative to discrimination power and prognostication following resection of HCC. (J Surg Oncol. 2022 Oct 4. doi: 10.1002/jso.27116.)

The advantage of our study is simplicity. Whereas Lima et al. used a more complicated point system (TBS low/medium/high = 0/1/2; AFP low/high = 0/1; Child Pugh class A/B = 0/1, respectively) to develop a risk score. We believe that model simplicity is of paramount importance in clinical application. Although numerous prognostic models have been developed for HCC (Eur J Surg Oncol. 2022;48:492-499. doi: 10.1016/j.ejso.2021.09.012.), the BCLC (J Hepatol 2022;76:681-693. doi: 10.1016/j.jhep.2021.11.018.) and the American Joint Committee on Cancer (AJCC) staging systems are the most popular, mainly due to their simplicity.

2.) What is the rational for using the Up-to-7 criteria in combination with AFP in the context of this analysis?

Response: To avoid confusing, we have deleted the relationship with up-to-seven criteria.

The results from Up-to-7 criteria in addition to AFP levels with regard to the prognostic stratification is not even mentioned in the abstract although the shown data is interesting. 

Response: To avoid confusing, we have deleted the relationship with up-to-seven criteria.

Reviewer 3 Report

This is a retrospective study evaluating the prediction ability of alpha-fetoprotein combined with radiographic tumor burden score. I have several comments.

1. The calculation method of tumor burden score should be demonstrated in abstract.

2. How is the final model of prediction? The relationship with up-to-seven criteria is confusing.

Author Response

This is a retrospective study evaluating the prediction ability of alpha-fetoprotein combined with radiographic tumor burden score. I have several comments.

  1. The calculation method of tumor burden score should be demonstrated in abstract.

Response: Thank you so much for your comments. We have demonstrated the calculation method of tumor burden score in abstract in the revised version.

  1. How is the final model of prediction? The relationship with up-to-seven criteria is confusing.

Response: To avoid confusing, we have deleted the relationship with up-to-seven criteria.

Round 2

Reviewer 1 Report

Congratulations for your work, however, I cannot find sufficient new evidence to published.

Author Response

Response: Thank you so much for your comments.

The TBS is simple to use, and numerous studies have demonstrated its predictive value in prognosis in patients with HCC under-going LR [6,7,23-32]. Among these studies, some studies used pathological TBS, which is not suitable for preoperative prognostic predictions[6,7,23-26]. Two studies only enrolled patients with BCLC stage B [27,28]. Whether the results of these studies could be extrapolated to other BCLC stages is unknown. Whereas our study enrolled patients with BCLC stage 0, A and B. Endo et al. reported that a preoperative model composed of AFP, radiographic TBS and neutrophil-to-lymphocyte ratio could predict the presence of microvascular invasion [29].Whereas the aim of the present study is to predict OS. Fukami et al. enrolled patients with BCLC stage 0, A, and B. The authors used the combination of the Controlling Nutritional Status score and radiographic TBS to predict OS. However, this study only enrolled 96 patients. The case number may be too small to draw the conclusion [30].

Endo et al. enrolled 1676 patients with BCLC stage 0, A, B and C. The authors used a preoperative model composed of radiographic TBS, AFP, neutrophil-to lymphocyte ratio, albumin, gamma-glutamyl transpeptidase, and vascular involvement to predict 5-year OS. This model could stratify OS into three distinct groups. Further, this model outperformed HCC staging systems including BCLC and the AJCC systems [31]. Lima et al. enrolled 1435 patients with HCC undergoing LR without mentioned their BCLC stage. A risk score which included three variables (i.e., radiographic TBS, AFP and Child-Pugh class) demonstrated superior predictive value for OS compared with BCLC stage and further stratified patients within BCLC stage relative to OS [32].

The advantage of our model is simplicity which is composed of radiographic TBS (cutoff value=7.9) and AFP (cutoff value=400 ng/ml). Whereas Lima et al. used a point system (TBS low/medium/high = 0/1/2; AFP low/high = 0/1; Child Pugh class A/B = 0/1, respectively) to develop a risk score [32]. Endo et al. used an online calculator (https://yutaka-endo.shinyapps.io/PrepoScore_Shiny/) for their complex model [31]. We believe that model simplicity is of paramount importance in clinical application. Although numerous prognostic models have been developed for HCC [33], the BCLC [5]and the AJCC [10,11 ] staging systems are the most popular, mainly due to their simplicity. Please see line 224-245.

Reviewer 3 Report

The authors have revised the manuscript appropriately.

Author Response

Response: Thank you so much for your comments.
